# Transfer-RNA-Derived Fragments Are Potential Prognostic Factors in Patients with Squamous Cell Carcinoma of the Head and Neck

**DOI:** 10.3390/genes11111344

**Published:** 2020-11-13

**Authors:** Xiaolian Gu, Lixiao Wang, Philip J. Coates, Linda Boldrup, Robin Fåhraeus, Torben Wilms, Nicola Sgaramella, Karin Nylander

**Affiliations:** 1Department of Medical Biosciences/Pathology, Umeå University, 90185 Umeå, Sweden; lixiao.wang@umu.se (L.W.); linda.boldrup@umu.se (L.B.); robin.fahraeus@inserm.fr (R.F.); sgaramellanicola12@gmail.com (N.S.); karin.nylander@umu.se (K.N.); 2Regional Centre for Applied Molecular Oncology (RECAMO), Masaryk Memorial Cancer Institute, 65653 Brno, Czech Republic; philip.coates@mou.cz; 3Institute of Molecular Genetics, University Paris 7, St. Louis Hospital, 75010 Paris, France; 4Department of Clinical Sciences/ENT, Umeå University, 90185 Umeå, Sweden; torben.wilms@umu.se

**Keywords:** tRNA-derived fragment, SCCHN, prognostic marker

## Abstract

Transfer-RNA-derived fragments (tRFs) are a class of small non-coding RNAs that are functionally different from their parental transfer RNAs (tRNAs). tRFs can regulate gene expression by several mechanisms, and are involved in a variety of pathological processes. Here, we aimed at understanding the composition and abundance of tRFs in squamous cell carcinoma of the head and neck (SCCHN), and evaluated the potential of tRFs as prognostic markers in this cancer type. We obtained tRF expression data from The Cancer Genome Atlas (TCGA) HNSC cohort (523 patients) using MINTbase v2.0, and correlated to available TCGA clinical data. RNA-binding proteins were predicted according to the calculated Position Weight Matrix (PWM) score from the RNA-Binding Protein DataBase (RBPDB). A total of 10,158 tRFs were retrieved and a high diversity in expression levels was seen. Fifteen tRFs were found to be significantly associated with overall survival (Kaplan-Meier survival analysis, log rank test *p*-value < 0.01). The top prognostic marker, tRF-20-S998LO9D (*p* < 0.001), was further measured in tumor and tumor-free samples from 16 patients with squamous cell carcinoma of the oral tongue and 12 healthy controls, and was significantly upregulated in tumor compared to matched tumor-free tongue (*p* < 0.001). Results suggest that tRFs are useful prognostic markers in SCCHN.

## 1. Introduction

Transfer-RNA-derived fragments (tRFs) are small non-coding RNAs with emerging functions in regulating gene expression [1,2,3,4,5]. In human cells, tRFs are produced constitutively or in response to a variety of stress stimuli [6,7,8]. Based on the cleavage position of tRNAs, tRFs are generally classified into six subtypes, with lengths of 14–50 nucleotides [7,9,10]. The endoribonuclease RNase Z processes a precursor tRNA to generate tRF-1 (also known as tsRNA), and angiogenin cleaves mature tRNAs within the anticodon loop to generate 5′-tRNA halves and 3′-tRNA halves. Other tRFs generated from the 5′ or 3′ ends or the internal region of mature tRNA, probably by the action of angiogenin or Dicer, are called 5′-tRFs, 3′-tRFs or i-tRFs, respectively [7,8,10]. tRNA halves are typically 33 nucleotides in length, whereas other tRFs are of variable length [11,12].

tRFs were first observed in 1977 in human tumor tissue [13] and suggested as markers of cancer [14]. Today, more than 20,000 tRFs are reported [12]. It is clear that the production of tRFs is associated with several factors such as gender, population origin, ethnicity, tissue, tissue state and disease subtype [7,15]. By regulating mRNA stability or protein translation initiation/elongation [16], tRFs are functionally involved in mammalian development [8], metabolic disorders [17], neurological disorders [18] and oncogenesis [9,10]. Dysregulation of tRFs has been shown in several cancers [15,19,20,21], exerting either tumor promoting [22] or tumor suppressive activities [23,24]. 

Squamous cell carcinoma of the head and neck (SCCHN) represents a heterogeneous group of tumors arising from squamous epithelium of the oral cavity, oropharynx, larynx and hypopharynx [25]. Human papilloma virus (HPV) infection, smoking and alcohol consumption are risk factors for development of SCCHN [25,26]. Today the molecular mechanisms underlying development and progression of SCCHN are better understood [26,27,28] and as many as 78 protein, DNA or mRNA prognostic biomarkers have been suggested, such as tumor-infiltrating lymphocytes, somatic copy number variations, and hypoxia gene signatures [29]. Being a group of small non-coding RNAs, which are stably enriched in various biofluids and tissues, the great potential of tRFs as cancer biomarkers has been recognized [30]. However, in SCCHN, even though a small number of tRFs have been linked to tumorigenesis [31,32], their impact on clinical outcome has not been investigated. 

The MINTbase v2.0 is a web-accessible repository comprising tRFs found in multiple human tissues [12]. According to the MINTmap algorithm [11], a total of 23,413 mature-tRNA-derived fragments with an abundance of ≥1.0 reads-per-million (RPM) were identified by processing 11,198 short RNA sequencing datasets from The Cancer Genome Atlas (TCGA). Due to the technical limitation of the short RNA sequencing method used by TCGA, tRFs longer than 30 nucleotides (e.g., tRNA halves) are under-represented among the TCGA datasets [12]. Nevertheless, MINTbase v2.0 provides the most comprehensive tRF expression data in 32 cancer types, including SCCHN [12]. 

In this study, we set out to explore the MINTbase v2.0 collected tRF data to identify the composition and abundance of tRFs in the TCGA head and neck squamous cancer (HNSC) cohort and to evaluate their potential use as prognostic markers. 

## 2. Materials and Methods 

### 2.1. Retrieval of tRF Data from MINTbase

The interactive database MINTbase v2.0 was employed to retrieve tRF data from the TCGA HNSC cohort. GRCh37/hg19 genome assembly was applied by MINTbase. MINTbase unique ID, which is introduced according to the “license plates” naming scheme, is used for tRF nomenclature. By definition, tRFs with identical sequence were given the same MINTbase unique ID, although they might map to multiple genomic locations. 

### 2.2. Identification of Survival-Associated tRFs

Clinical data of the TCGA HNSC cohort was downloaded from TCGA’s data portal https://portal.gdc.cancer.gov/. HPV status was downloaded from the Firehose database http://firebrowse.org/. In order to obtain higher statistical power, we focused on tRFs that were present in at least 100 patients. Kaplan–Meier survival analysis [33] was conducted by using R survival and survminer packages [34]. Patients were classified into high and low groups according to the median tRF level of tumor samples. A log-rank test with *p*-value < 0.01 was considered statistically significant. 

### 2.3. Prediction of tRF Binding Proteins

tRFs can interact with RNA-binding proteins (RPBs) and regulate their function [23,35,36]. To estimate the role of tRFs in SCCHN, we used the RNA-Binding Protein DataBase (RBPDB) [37] to predict tRF binding proteins. The submitted tRF sequences were scanned with a set of Position Weight Matrix (PWM) in the database. Every site that scored at least 80% of the maximum possible score for that matrix was returned. The predicted *Homo sapiens* proteins were considered potential tRF binding proteins. 

### 2.4. Relationship between tRF Level and Clinical Features

The top survival-associated tRF was selected to explore the relationship between gene expression and clinical features. Associations between categorized clinical variables and categorized tRF levels were determined by Chi-Square test. The nonparametric Mann–Whitney U test was used to study the difference between two groups of continuous variables. In multivariate Cox regression analysis, patient age at diagnosis, gender, HPV status and clinical stage were considered as covariates. Statistical tests were conducted in IBM SPSS Statistics 26 (IBM Corp., Armonk, NY, USA). A two-sided *p*-value < 0.05 was considered statistically significant.

### 2.5. Quantification of tRF Levels Using Reverse Transcription Quantitative PCR (RT-qPCR)

For the top survival-associated tRF, RT-qPCR was performed to measure expression levels in our clinical samples: 12 healthy volunteers and 16 patients with squamous cell carcinoma of the oral tongue (SCCOT) (Appendix A). The healthy volunteers and patients were not matched for gender and age, as the number of healthy individuals willing to donate tongue tissue was limited. As described previously [38,39], for healthy volunteers not exposed to classical oral cancer risk factors (smoking and alcohol), biopsies were taken from the lateral border of the tongue. For patients with SCCOT, biopsies were taken from the tumor, as well as clinically normal tongue contralateral to the tumor. The study was approved by the Regional Ethics Review Board, Umeå, Sweden (Dnr 08-003 M) and performed in accordance with the Declaration of Helsinki. Written informed consent was obtained from all participants. 

TaqMan™ small RNA assays were selected for tRF quantification (Thermo Fisher Scientific, Waltham, MA, USA). tRF-specific reverse transcription primers and PCR primers were designed and produced by the company and the U6 TaqMan™ Small RNA Control was used as endogenous control. According to the manufacturer’s protocol, tRF-specific cDNA was synthesized from 7.5 ng of total RNA. RT-qPCR was performed using the QuantStudio 6 Flex real-time PCR system with TaqMan™ Fast Advanced Master Mix (Thermo Fisher Scientific). The 2^−ΔΔCT^ method was applied to calculate relative gene expression levels. Using SPSS Statistics 26, nonparametric Mann–Whitney U test or Wilcoxon signed-rank test were carried out to compare the difference between healthy controls and tumor-free samples, or between matched SCCOT and tumor-free samples, respectively.

## 3. Results

### 3.1. Reported tRFs in the TCGA HNSC Cohort

TCGA has studied 523 patients with SCCHN using short RNA sequencing, including 525 tumor samples (523 primary and 2 metastatic) and 44 normal controls. In the MINTbase v2.0 database, expression of 10,158 tRFs were recorded for this cohort, representing 43% of all tRFs reported from all 32 TCGA cancer cohorts. Expression values of 2436 tRFs were reported only in single samples, 3194 in 2 to 9 samples, 2701 tRFs in 10 to 99 samples and 1827 tRFs in 100 to 569 samples. Levels of tRFs ranged from 1.002 RPM to 61,097.254 RPM. The maximum expression level was lower than 10 RPM for 7473 of 10,158 tRFs (74%).

According to MINTbase v2.0, the 10,158 tRFs might be derived from 549 tRNAs and from 51,755 tRNA genomic loci. As shown in Figure 1A, both mitochondrial (MT) and nuclear tRFs were identified and the majority of tRFs were mapped to chromosomes 1 and 6. Most tRFs were derived from tRNAs for glutamic acid (Glu/E) (Figure 1B). The length of tRFs ranged from 16 to 30 nucleotides (Figure 1C). The i-tRFs comprise the most abundant class of tRFs (71%, 7183/10158), followed by 3′-tRFs (17%) and 5′-tRFs (12%). Only 23 tRFs belonged to the 5′-half class and no 3′-half tRFs were identified (Figure 1D). 

### 3.2. Prognosis-Associated tRFs in SCCHN

To identify prognosis-associated factors, the 1827 tRFs reported in at least 100 samples were investigated. Among these, 53% belonged to the i-tRF class, 25% to the 5′-tRFs and 21% to the 3′-tRF. There were only nine 5′-tRNA halves, all of which were derived from mitochondrial tRNAs. Patients were classified into high- and low-level groups according to the median tRF level of tumor samples. Kaplan–Meier overall survival analysis using the Survminer R packages revealed that significant differences between high- and low-level groups were seen for 15 tRFs (log-rank test, *p* < 0.01, Table 1). Detailed information for the 15 tRFs, which are mapped to 59 genomic locations, is listed in Appendix A.

### 3.3. tRF-Interacting Proteins

The 15 tRFs with significant impact on survival were scanned through the RNA-Binding Protein DataBase (RBPDB). As shown in Table 1, 11 different RNA binding proteins were predicted. Six proteins are RNA splicing factors (SRSF1, SRSF9, SRSF10, RBMX, FUS, KHDRBS3), four are splicing regulators (YTHDC1, NONO, YBX1, YTHDC2) and one a translation initiation factor (eIF4B). 

### 3.4. The Association between tRF-20-S998LO9D and Clinical Factors

As shown in Table 1, according to the *p*-value of the Kaplan–Meier log rank test, tRF-20-S998LO9D, tRF-16-I8W47WB and tRF-16-884U1DD are the top three tRFs associated with patient survival (*p* = 0.003). Compared to the other two tRFs, expression tRF-20-S998LO9D, hereafter referred to as tRF-20, was reported in more tumor samples (443 primary SCCHN tumors, 1 metastatic sample and 16 normal controls). Therefore, we considered tRF-20 as the top-ranked tRF associated with patient survival. tRF-20 is classified as a 5′-tRF and exclusively derived from chromosome 1 tRNA86^ArgTCT^. According to RBPDB prediction, this fragment binds to the translation initiation factor eIF4B. Based on these findings, we decided to further explore the relationship between tRF-20 and clinical features. Patients were divided into three groups based on log^2^-transformed RPM levels (less than 2, 2 to 4, higher than 4), as shown in Appendix A. The Chi-square test indicated that there was no significant correlation between tRF-20 levels and gender, age, clinical stage or HPV status (*p* > 0.05). When comparing the 443 primary tumors to the 16 normal controls, significantly higher tRF-20 levels were seen in tumor samples (*p* < 0.001) (Figure 2A). We also investigated tRF-20 expression in different subtypes of SCCHN. As shown in Appendix A, the top two tumor subsites demonstrating tRF-20 expression were tongue (*n* = 113) and larynx (*n* = 99). No significant difference in tRF-20 levels was seen between tongue and larynx cancer (*p* = 0.896). Expression of tRF-20 was found in 64 tumors in the mixed group of “overlapping lesion of lip, oral cavity and pharynx”, with significantly higher levels compared to both tongue (*p* = 0.003) and laryngeal (*p* = 0.001) SCC.

Kaplan–Meier survival plot indicated that patients with high tRF-20 have poorer overall survival than patients with low levels of tRF-20 (Figure 2B). We further studied the impact of tRF-20 using multivariate Cox-regression analysis. Considering gender, age at diagnosis, HPV infection and clinical stage as co-variates, tRF-20 remained significantly associated with overall survival (*p* = 0.002, HR = 1.624, 95% CI = 1.213 to 2.219). ROC curve revealed that the specificity of tRF-20 in predicting patient overall survival is 0.614, and the sensitivity is 0.643 (AUC = 0.669, *p* < 0.001, Appendix A).

### 3.5. Investigation of tRF-20 in Our Clinical Samples

The differential expression of tRF-20 was validated in a small cohort of patients with SCCOT. In agreement with the TCGA SCCHN cohort, tRF-20 was significantly upregulated in SCCOT samples compared to the matched tumor-free tongue tissue (*p* < 0.001, Figure 2C). Comparing tumor-free samples to healthy controls, no significant difference was found (*p* = 0.963, Figure 2C).

## 4. Discussion

Although several studies have described dysregulation of tRFs in cancer, their impact on patient survival remains poorly investigated. Here, we investigated the composition and abundance of tRFs in SCCHN, as well as their prognostic potential. 

The MINTbase v2.0 is a web-accessible repository providing the most comprehensive tRF expression data in 32 TCGA cancer types, including SCCHN. It provides users with access to a large collection of tRFs, which were identified through a deterministic and exhaustive tRF mining pipeline [12]. However, so far, the MINTbase v2.0 only comprises mature tRNA derived fragments with an abundance of ≥1.0 RPM, meaning that if expression was not recorded, levels could be either 0 or just under the threshold. Due to the overall low read-count of tRFs and the low number of normal controls available, this introduces a problem when calculating differential expression between tumor and normal samples. To avoid this problem, we aimed at identifying tRFs related to prognosis in this study. Among 10,158 detected tRFs in SCCHN, only 1827 were identified in at least 100 samples. Together with the large range in tRF levels among samples, it is obvious that tRF expression is highly diverse and probably varies between cell types and cellular states [7,36]. Nevertheless, according to Kaplan–Meier analysis, 15 tRFs were potential prognostic markers for patients with SCCHN.

Previously, differential expression of multiple 5′ tRNA halves had been shown in serum and/or tumor tissue from SCCHN patients as compared to control counterparts, indicating the potential of 5′ tRNA halves as diagnostic biomarkers for SCCHN [31,32]. However, in this study, among 10,158 MINTbase recorded tRFs for SCCHN, there were only 23 tRFs belonging to the class of 5′ tRNA halves. Furthermore, no prognostic value of 5′ tRNA halves for SCCHN was seen. Due to the technical limitation of TCGA data using short RNA sequencing, only fragments shorter than 30 nt could be reliably identified. Therefore, the contribution of long tRFs, such as tRNA halves that are produced under stress and involved in translation regulation [35,40,41], is likely to be underestimated in this study. The specificity and sensitivity of tRF-20 are not sufficient for predicting patient survival. Still, we could demonstrate a prognostic potential of tRF-20, and the performance of tRF-20 could be improved when a multiple biomarker panel is used. In clinical practice classical prognostic factors such as anatomic location, tumor size (T stage), nodal involvement (N stage), distant metastases (M stage), tumor grading, treatment and resection margins are widely used [29,42]. Of the many new prognostic biomarkers human papillomavirus status is the most robust prognostic factor for survival among patients with oropharyngeal SCC [29,43,44]. However, sensitivity and specificity of these markers have not been reported. There is thus an unmet need for statistical evaluation of markers with respect to sensitivity and specificity.

tRFs can bind to RNA or protein and affect mRNA translation [10]. To estimate the functions of the prognostic tRFs we found, the RBPDB database was used to predict RNA interacting proteins. Without validation, these results could be considered non-significant. However, we show a common feature of the predicted proteins being related to RNA splicing, except for eIF4B (eukaryotic translation initiation factor 4B). We found that eIF4B was predicted to interact with three tRFs. Previous studies have shown that eIF4B controls cell survival and proliferation and is regulated by oncogenic signaling pathways [45,46]. eIF4B is a cofactor of the eukaryotic initiation factor 4A (eIF4A), a subunit of the heterotrimeric cap-binding complex eIF4F. It has been reported that 5′-tRNA halves can bind directly or indirectly to eIF4G/eIF4A and reduce the rate of translation initiation [35]. However, interactions between 5′-tRFs and translation initiation factors have not yet been identified, even though some 5′-tRFs inhibit protein translation [47]. Under certain stress conditions, a 26 nt long 5′-tRF produced from valine-tRNA competed with mRNA for ribosome binding, resulting in global translation attenuation [2]. 

For the rest of the predicted tRF binding proteins, YBX1 (Y-Box Binding Protein 1) is the only one for which an interaction with specific tRFs has been previously shown. Goodarzi et al. reported that binding of hypoxic stress induced tRFs to this oncogenic RNA-binding protein led to destabilization of pro-oncogenic transcripts and thereafter suppresses the development of breast cancer metastasis [23]. RBMX (RNA Binding Motif Protein X-Linked), NONO (Non-POU Domain Containing Octamer Binding) and FUS (FUS RNA Binding Protein) are the three RNA-binding proteins that have been demonstrated to be potential cancer drivers [48]. The role of FUS as a tumor suppressor was seen in bladder urothelial carcinoma and SCCHN [48]. 

Due to limited and uneven sample size and incomplete clinical information (e.g., primary treatment) of the TCGA-HNSC cohort, the REMARK (Reporting Recommendations for Tumour Marker Prognostic Studies) guidelines [49] could not be completely followed in this study. The prognostic value of tRFs should therefore be further investigated in another cohort with sufficient patients and clinical data. According to TCGA data, significantly higher tRF-20 levels were seen in 443 tumor samples compared to 16 normal controls. As the number of tumor samples and normal controls is highly unbalanced, variability of tRF expression in the normal tissue could not be fully addressed. Nevertheless, using a small number of clinical samples, we successfully quantified the level of tRF-20 using RT-qPCR, and demonstrated that tRF-20 was significantly up-regulated in SCCOT compared to matched tumor-free tongue. Using our clinical samples, we found that there was no significant difference in tRF-20 expression between healthy controls and tumor-free samples. However, this result was limited due to the fact that the 12 healthy controls and 16 patients were not gender- and age-matched. TCGA data showed that the top two tumor subsites demonstrating tRF-20 expression was tongue and larynx. Due to the lack of laryngeal samples in our biobank, we could not verify expression of tRFs in this tumor subtype. Finally, as the number of our patients with tumors at different stages was small, no analysis of the impact of tRF-20 expression on patient survival was performed.

## 5. Conclusions

Analysis of TCGA data showed that tRFs act as prognostic markers in several different sub-sites of SCCHN. The top prognostic marker, tRF-20, emerged as a promising clinical biomarker and its upregulation in tumor was demonstrated in an independent group of patients with SCCOT.

## Figures and Tables

**Figure 1 genes-11-01344-f001:**
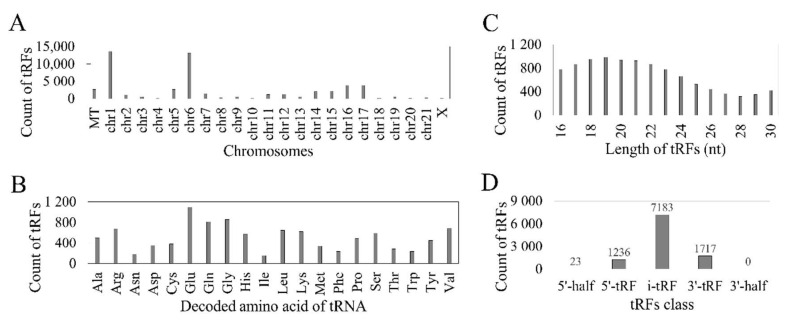
Overview of reported tRFs in the TCGA HNSC cohort. (**A**) Potential mapping of tRFs at mitochondria (MT) and different chromosomes. (**B**) Corresponding tRNA decoded amino acids (three letter code). (**C**) Length-wise distribution of tRFs. (**D**) Count of fragments according to the five tRF classes.

**Figure 2 genes-11-01344-f002:**
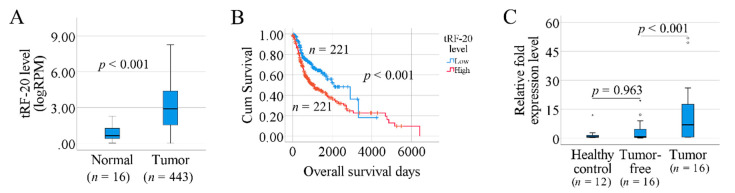
Investigation of tRF-20 in SCCHN. (**A**) Box-plots of tRF-20 levels in 16 normal and 443 tumor samples (TCGA data). (**B**) Kaplan–Meier curves showing the impact of tRF-20 levels on overall survival of patients with SCCHN (TCGA). (**C**) Box-plots showing relative fold expression levels of tRF-20 in 12 healthy controls and 16 pairs of tumor-free/tumor samples from patients with SCCOT (our clinical samples). Small circles indicate outliers and asterisks indicate extreme outliers.

**Table 1 genes-11-01344-t001:** Prognostic tRFs in SCCHN.

MINTbase Unique ID (Sequence Derived)	Fragment Sequence	*p*-Value (Kaplan-Meier log Rank Test)	Number of Tumor	Number of Normal	Average Level in Tumor (RPM)	Average Level in Normal (RPM)	Chr	Type	Amino Acid and Anticodon	Predicted RNA Binding Protein
tRF-20-S998LO9D	GTCTCTGTGGCGCAATGGAC	0.0003	443	16	20	2	1	5′-tRF	ArgTCT	eIF4B, SRSF1
tRF-16-I8W47WB	ATTGGTCGTGGTTGTA	0.0003	285	37	12	29	MT	i-tRF	GluTTC	
tRF-16-884U1DD	TCCGGCTCGAAGGACC	0.0003	255	37	10	8	14	3′-tRF	TyrGTA	SRSF9, eIF4B, SRSF1
tRF-22-8XF6RE98N	TCCTAAGCCAGGGATTGTGGGT	0.0006	466	42	6	11	16	i-tRF	ArgCCT	NONO, RBMX
tRF-21-NYDRFU8U0	CTTTGAATCCAGCGATCCGAG	0.0011	340	30	3	3	6	i-tRF	GlnTTG	YTHDC2, RBMX
tRF-21-I8W47W1R0	ATTGGTCGTGGTTGTAGTCCG	0.0024	507	41	31	49	MT	i-tRF	GluTTC	
tRF-21-LE3JWB61B	CGAATCCGGCTCGAAGGACCA	0.0030	209	29	3	2	6	3′-tRF	TyrGTA	SRSF9, YTHDC1, eIF4B, SRSF1
tRF-20-6S7P4PWJ	GGCCGGTTAGCTCAGTCGGC	0.0031	303	14	7	2	6	5′-tRF	IleAAT	
tRF-23-Z87HFK8SDZ	TTTGGGTGCGAGAGGTCCCGGGT	0.0039	375	30	4	2	14	i-tRF	ProTGG	FUS, RBMX, SRSF10
tRF-23-H3RXSINH0P	ATAGTGGTTAGTACTCTGCGTTG	0.0050	177	8	5	5	1	i-tRF	HisGTG	YBX1, YTHDC1
tRF-19-Z8SSFKJJ	TTTGGGTCCGAGAGGTCCC	0.0063	155	19	2	2	11	i-tRF	ProTGG	SRSF10
tRF-19-Q99P9PJZ	GCTTCTGTAGTGTAGTGGT	0.0063	260	19	4	4	6	5′-tRF	ValCAC	
tRF-20-MEF91SS2	CGGATAGCTCAGTCGGTAGA	0.0069	115	26	2	2	11	i-tRF	LysTTT	
tRF-30-XSXMSL73VL4Y	TGCCGTGATCGTATAGTGGTTAGTACTCTG	0.0074	388	27	10	3	1	5′-tRF	HisGTG	YTHDC1
tRF-21-7OFIZ9WUD	GTTAAAGACTTTTTCTCTGAC	0.0075	204	8	4	2	MT	3′-tRF	ProTGG	SRSF10, KHDRBS3

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
