# Peer review of "Transfer-RNA-Derived Fragments Are Potential Prognostic Factors in Patients with Squamous Cell Carcinoma of the Head and Neck"

_genes, 2020, doi:10.3390/genes11111344_

Round 1
Reviewer 1 Report
In the manuscript entitled "Transfer-RNA-derived fragments are prognostic factors in patients with squamous cell carcinoma of the head and neck" by Xiaolian Gu et. al, the authors analyze the composition and abundance of Transfer-RNA-derived fragments (tRFs) in squamous cell carcinoma of the head and neck (SCCHN). They show fifteen tRFs which are significantly associated with overall survival. Furthermore, they predict proteins that can bind mentioned tRFs. Finally, the authors indicate tRF-20-S998LO9D as a useful prognostic marker in SCCHN.
The work by Xiaolian Gu et. al, can be important in the field. However, the manuscript should be more comprehensive.
Major concerns:
- In the introduction, the authors should report other prognostic markers that were proposed in SCCHN. They should also point out how tRFs can be better.
- lines 139-140: it is not clear what does "low-level group" mean?
- It should be specified why tRF-20-S998LO9D is the top-ranked tRF associated with patient survival (besides being expressed in 460 samples)
- Prediction of proteins that can bind tRFs is not enough. The authors should verify at least binding to tRF-20-S998LO9D
- Specify, please, what is the specificity and sensitivity of tRF-20-S998LO9D as a prognostic marker
- In the discussion, the authors should refer to recommendations for tumor marker prognostic studies (REMARK) (Altman et al. BMC Medicine 2012) and answer the question of whether tRF-20-S998LO9D fulfill all the criteria of the prognostic marker.
Major concerns:
- Please, format table 1 column description
Reviewer 2 Report
With this work, the authors aimed to investigate the role of tRFs as a prognostic markers in head and neck squamous cell carcinoma (HNSCC). They conducted the study by obtaining tRF expression data from TCGA cancer patients using interactive MINTbase. Also the RNA-Binding Protein Database was used for identification of proteins which may interact with selected tRFs. The authors demonstrated the involvement of fifteen tRFs significantly associated with overall survival of HNSCC. The most promising prognostic factor was further validated on small group of patients with squamous cell carcinoma of tongue. The work is interesting, the manuscript is well written and comprehensible, but the authors should answer some important questions, in order to strengthened the results.
Specific comments:
Abstract:
- The number of samples should be added. There is no information about healthy controls group and number of matched tumor-free tongues samples.
- The information about analysis with RNA-Binding Protein Database should be added.
Materials and methods:
- There is no any information about 12 healthy volunteers samples. If the samples were matched by gender, age, alcohol/cigarette consumption status? From supplementary data it seems they didn't. Authors should add the explanation of that and describe this in limitation of the study section.
- In addition, the number of control samples, presented in Table 2 as “normal” is quite small in comparison to number of cancer patients samples. It also should be add to the limitation of the study section.
Results:
- In the presentation of the results the authors must include the number of samples on plots.
- The sentence “own clinical samples” should be changed (“own” deletion).
- As the most significant results were obtained for larynx and tongue subtypes of HNSCC samples, the validation should aimed the larynx samples too. Authors should explained if this avoidance is a result of larynx samples absence?
Discussion:
- Previously, profiling of small non-coding RNAs among head and neck squamous cell carcinoma (HNSCC) indicated that 5′ tRNA-Val-CAC-2-1 half was reduced in serum and tumor tissue of the patients. Authors should consider this data in the discussion section. There is no any reference to either Dhabibi J. nor Martinez J. results in discussion.
- In section connected with RBPDB database, there is only discussion about one protein eIF4B. This section should be completed with discussion of other proteins identified during this study. Is one of the protein was previously associated with HNSCC?
- The authors might explain and discuss the advantages and disadvantages of chosen databases.
Round 2
Reviewer 1 Report
The revised manuscript entitled "Transfer-RNA-derived fragments are prognostic factors in patients with squamous cell carcinoma of the head and neck" by Xiaolian Gu et. al, has been improved, however, there are still a few concerns that should be addressed.
- The manuscript title is misleading. The authors did not demonstrate that tRF-20 can be prognostic marker. They admitted that the survey had some limitations and "prognostic value of tRFs should be further investigated"
- The specificity of tRF-20 0.614, and the sensitivity 0.643 is not overwhelming. How does it look like in case of other SCCHN prognostic markers?
- The results presented in “tRF-interacting proteins” section are insufficient and insignificant without validation
Author Response
The revised manuscript entitled "Transfer-RNA-derived fragments are prognostic factors in patients with squamous cell carcinoma of the head and neck" by Xiaolian Gu et. al, has been improved, however, there are still a few concerns that should be addressed.
- The manuscript title is misleading. The authors did not demonstrate that tRF-20 can be prognostic marker. They admitted that the survey had some limitations and "prognostic value of tRFs should be further investigated"
Reply: Thanks for your comment! The title is now changed to “Transfer-RNA-derived fragments are potential prognostic factors in patients with squamous cell carcinoma of the head and neck”, as shown at line 2.
- The specificity of tRF-20 0.614, and the sensitivity 0.643 is not overwhelming. How does it look like in case of other SCCHN prognostic markers?
Reply: We fully agree that the specificity and sensitivity of tRF-20 are not sufficient for predicting patient survival. Still, we could demonstrate a prognostic potential of tRF-20, and the performance of tRF-20 could be improved when a multiple biomarker panel is used. In clinical practice classical prognostic factors such as anatomic location, tumor size (T stage), nodal involvement (N stage), distant metastases (M stage), tumor grading, treatment and resection margins are widely used. Of the many new prognostic biomarkers human papillomavirus status is the most robust prognostic factor for survival among patients with oropharyngeal SCC. However, sensitivity and specificity of these markers have not been reported. There is thus an unmet need for statistical evaluation of markers with respect to sensitivity and specificity. This is now added in the Discussion, line 224-232.
- The results presented in “tRF-interacting proteins” section are insufficient and insignificant without validation.
Reply: On lines 235-237 we have now added: Without validation, these results could be considered non-significant. However, we showed a common feature of the predicted proteins being related to RNA splicing, except for eIF4B.